# Optimization of a Novel Tyrosinase Inhibitory Peptide from *Atrina pectinata* Mantle and Its Molecular Inhibitory Mechanism

**DOI:** 10.3390/foods12213884

**Published:** 2023-10-24

**Authors:** Wen Wang, Haisheng Lin, Weiqiang Shen, Xiaoming Qin, Jialong Gao, Wenhong Cao, Huina Zheng, Zhongqin Chen, Zhishu Zhang

**Affiliations:** 1College of Food Science and Technology, Guangdong Ocean University, Zhanjiang 524088, China; wangwen31@stu.gdou.edu.cn (W.W.); qinxm@gdou.edu.cn (X.Q.); gaojl@gdou.edu.cn (J.G.); cwenhong@gdou.edu.cn (W.C.); zhenghn@gdou.edu.cn (H.Z.); chenzhongqin@gdou.edu.cn (Z.C.);; 2National Research and Development Branch Center for Shellfish Processing (Zhanjiang), Zhanjiang 524088, China; 3Guangdong Provincial Key Laboratory of Aquatic Products Processing and Safety, Zhanjiang 524088, China; 4Guangdong Provincial Engineering Technology Research Center of Seafood, Zhanjiang 524088, China; 5Guangdong Province Engineering Laboratory for Marine Biological Products, Zhanjiang 524088, China; 6Shenzhen Institute of Guangdong Ocean University, Shenzhen 518108, China; 7Collaborative Innovation Center of Seafood Deep Processing, Dalian Polytechnic University, Dalian 116034, China

**Keywords:** *Atrina pectinata* mantle, gelatin peptide, inhibition kinetic, molecular docking

## Abstract

In order to realize the multi-level utilization of marine shellfish resources and to develop the potential biological activity of processing by-products of *Atrina pectinata*, gelatin was extracted from the mantle and the potential whitening effect of its enzymatic peptides was explored. Taking tyrosinase inhibitory activity as the evaluation index, the enzyme hydrolysate process was optimized by response-surface methodology, and the optimal enzyme hydrolysate conditions were as follows: pH 5.82, 238 min enzyme hydrolysate time, and temperature of 54.5 °C. Under these conditions, the tyrosinase inhibition activity of *Atrina pectinata* mantle gelatin peptide (APGP) was 88.6% (IC_50_ of 3.268 ± 0.048 mg/mL). The peptides obtained from the identification were separated by ultrafiltration and LC–MS/MS, and then four new peptides were screened by molecular docking, among which the peptide Tyr-Tyr-Pro (YYP) had the strongest inhibitory effect on tyrosinase with an IC_50_ value of 1.764 ± 0.025 mM. The molecular-docking results indicated that hydrogen bonding is the main driving force for the interaction of the peptide YYP with tyrosinase. From the Lineweaver–Burk analysis, it could be concluded that YYP is inhibitory to tyrosinase and exhibits a mixed mechanism of inhibition. These results suggest that YYP could be widely used as a tyrosinase inhibitor in whitening foods and pharmaceuticals.

## 1. Introduction

Melanin is synthesized by melanocytes in the vascular layer of the epidermis and is primarily responsible for the color of hair, eyes and skin. Melanin production helps protect the skin from ultraviolet (UV) damage, however, excess melanin can lead to epidermal hyperpigmentation and the risk of hyperpigmentation diseases such as melasma, freckles and age spots [1,2]. Tyrosinase, the key enzyme responsible for melanin biosynthesis, is located in the basal layer of the skin and is produced in specific dendritic cells of melanocytes, inducing melanin synthesis through various internal factors such as alpha-melanotropic hormone (a-MSH) and environmental factors such as UV light [3]. Tyrosinase catalyzes two different reactions for the synthesis of melanin: hydroxylation of monophenolic molecules to form o-phenols and oxidation of o-phenols to form o-quinones. O-quinone forms polymeric complexes under oxidation and polymerization reactions, eventually converting to melanin [4]. Therefore, inhibition of tyrosinase activity is an effective way to prevent and treat melanogenesis.

Chemicals with excellent whitening properties are mainly phenolic substances, including triglycerides, hydroquinone, arbutin and epicatechin [5,6,7]. Triglyceride represents the most promising class of tyrosinase inhibitor due to its chelating properties with metallic copper and its phenolic structure. However, the toxicity and chemical instability of triglyceride to normal skin fibroblasts limit its wide application in cosmetics [8]. Hydroquinone has been banned by the European Commission for use in cosmetics due to its potential mutagenicity to mammalian cells, and arbutin’s low lipophilicity may lead to poor skin permeability and low formulation stability [9]. Therefore, there is still a need to seek tyrosinase inhibitors with hydrophilic and lipophilic skin-whitening properties as a way to slow down the adverse reactions caused by the skin-penetration process.

Gelatin peptides are prepared from gelatin or collagen by acid–base methods, microbial fermentation, enzymatic methods, etc. They are generally composed of 2–20 amino acid residues and have specific functional activities [10]. The biological activities of gelatin peptides, such as antioxidant activity [11], angiotensin-converting enzyme (ACE) inhibitory activity [12], antibacterial activity [13], hypotensive activity [14,15] and promotion of osteoblast proliferation and mineralization [16], have been reported. Gelatin peptides extracted from natural sources are characterized by high skin compatibility and mild and easy absorption, and their amino acid residues are correlated with whitening activity. A recent study showed that the gelatin peptide extracted from fish scales exhibits 61.7% tyrosinase inhibition (5 mg/mL) [17]. The identified peptide, DLGFLARGF, binds amino acid residues around the D chain of tyrosinase and inhibits the enzyme activity through catalytic and metastable sites of hydrogen bonding and hydrophobic interactions. Li [18] demonstrated that silver carp scale collagen peptides (SCPs1) can reduce tyrosinase activity and reactive oxygen species (ROS) content of B16F10 mouse melanoma cells, thus affecting melanogenesis, and they are expected to be applied to skin-whitening products.

To the best of our knowledge, the possible effects of gelatin peptides from *Atrina pectinata* mantle on tyrosinase activity have not yet been described. *Atrina pectinata*, also known as pen shell in English, is a filter-feeding, marine, soft-bodied bivalve shellfish with high commercial value in the Asia–Pacific region. Its adductor muscle is popular in East Asia for its sweet taste and nutrients, and the remaining parts, for instance, the mantle and visceral tissues, are often discarded as processing waste [19]. Furthermore, most of the existing research has focused on aquaculture, genetic breeding and antioxidant enzyme systems, with less attention paid to multistage utilization of processing wastes and bioactive peptides [20,21]. Therefore, in the present study, tyrosinase inhibitory peptides were prepared from *Atrina pectinata* mantle, with the process being optimized by response-surface methodology. The peptides were then separated and identified, successively, by ultrafiltration, LC-MS/MS and molecular docking. Solid-phase synthesis of the screened inhibitory peptides and determination of IC_50_ values were performed, and the molecular inhibitory mechanism of the optimal tyrosinase inhibitory peptide was investigated for the rational utilization of the *Atrina pectinata* mantle. The aim of the study was to expand the use of the peptide as a natural skin whitener in the biomedical and cosmetic industries.

## 2. Materials and Methods

### 2.1. Materials

*Atrina pectinata* was purchased from the local Dongfeng Market in Zhanjiang, China, and the *Atrina pectinata* mantle gelatin (APG) was prepared as described previously [22]. Mushroom tyrosinase (500 U/mg) was purchased from Beijing Solarbio Biotechnology Co., Ltd. (Beijing, China). Trypsase (250 U/mg), alkaline protease (200 U/mg), papain (800 U/mg), animal protease (100 U/mg), neutral protease (100 U/mg) and L-tyrosine were procured from Shanghai Yuanye Biotechnology Co., Ltd. (Shanghai, China). Kojic acid (≥99%) was ordered from Shanghai Macklin Biotechnology Co., Ltd. (Shanghai, China). All the other reagents were analytical grade.

### 2.2. Preparation of Atrina pectinata Mantle Gelatin Peptides (APGPs)

The protein concentration of the APG solution was stabilized at 3.8 mg/mL. Alkaline protease, animal protease, neutral protease, trypsin and papain were selected for enzymatic hydrolysis under their respective optimal conditions, and the best protease was screened by using tyrosinase inhibition rate as an evaluation index. The effects of enzyme concentration (2, 3, 4, 5 and 6 U/mg), enzymatic pH (5, 6, 7 and 8), time (3, 3.5, 4 and 4.5 h) and temperature (45, 55, 65 and 75 °C) of this protease on the inhibition of tyrosinase and degree of hydrolysis (DH) were explored. After enzyme hydrolysis, the enzyme was inactivated in a boiling water bath for 15 min, restored at room temperature, centrifuged at 12,000 rpm/min at 4 ℃ for 15 min, and the supernatant was lyophilized as *Atrina pectinata* mantle gelatin peptides (APGPs).

### 2.3. Optimization of Enzymatic Parameters via Response-Surface Methodology (RSM)

According to the design principle of the Box–Behnken central-composite experiment, with the enzymolysis time (A), temperature (B) and pH (C) as independent variables, and tyrosinase inhibition rate as response value (Y), a response surface with three factors and three levels was designed to optimize the enzymolysis conditions.

### 2.4. Analysis of the Tyrosinase Inhibitory Activity

Tyrosinase inhibitory activity was determined in conformity with [23]. With reference to the reagent addition in Table 1, we added phosphate buffer solution (PBS, pH 6.8, 0.1 mol/L), sample solutions and L-tyrosine solution (L-Tyr, 0.5 mg/mL, pH 6.8 in PBS) to each well in a 96-well plate. These were mixed well and incubated at 37 °C for 10 min. Then, 20 μL of mushroom tyrosinase solution (500 U/mL) was added, mixed and reacted at 37 °C for 15 min, and monitored at 475 nm. The tyrosinase inhibition rate (*Y*) was calculated as follows:(1)$$Y=(1-\frac{A_{4}-A_{3}}{A_{2}-A_{1}})\times 100\%$$
where A_4_ is the absorbance of the reaction well of the sample, A_3_ is the absorbance value of the sample background well, A_2_ is the absorption value of the solvent reaction well and A_1_ is the absorbance of the solvent background well.

### 2.5. Assay of DH of APGP

The hydrolysis degree of APGP was determined on the basis of the method proposed by Bai [24]. The amount of total and non-protein nitrogen in the raw material was determined by the Kjeldahl method, and the amino nitrogen content of the hydrolysate and the free amino nitrogen content of the raw materials were determined by potentiometric titration of neutral formaldehyde. *DH* was calculated by Equation (2):(2)$$DH=\frac{A_{2}-A_{3}}{A_{1}-A_{4}}\times 100\%$$
where A_1_ is the amount of total nitrogen in the APG, A_2_ is amino nitrogen in the APGP, A_3_ is the content of free amino nitrogen in the APGP and A_4_ is the amount of non-protein nitrogen in the APG.

### 2.6. Examination of Amino Acids

We referred to and appropriately modified the methodology of Chen et al. [25] for the determination and analysis of amino acid composition. Compositional analysis of APGP was performed using an L-8900 high-speed amino acid analyzer (Hitachi Ltd., Tokyo, Japan), and the treatment conditions were as follows: hydrolysis under HCl (6 mmol/L) for 24 h (110 ℃).

### 2.7. Analysis of Molecular-Weight Distribution (MWD)

The molecular weight (MW) was distributed by an Agilent 1260 Infinity HPLC system (Agilent Technologies, Inc, Santa Clara, CA, USA) equipped with Waters Ultra Hydrogel 500-250-120a GPC (7.8 × 300 mm). APGP was eluted with 0.1 mol/L NaNO_3_ aqueous solution at a flow rate of 1 mL/min (injection volume: 40 μL), and the detection wavelength was 220 nm. The MW of the peptide was calculated according to the calibration curve constructed by polyethylene glycol (PEG) with different MWs.

### 2.8. Ultrafiltration Separation

Two kinds of ultrafiltration tubes with molecular weight cut-off (MWCO) of 5 and 3 kDa were used successively. Three kinds of ultrafiltration components with MW > 5 kDa, 3~5 kDa and <3 kDa were obtained. After freeze-drying, the tyrosinase inhibitory activity of each component was determined, and the component with the highest tyrosinase inhibitory activity was selected for the next experiment.

### 2.9. Identification of the Tyrosinase Inhibitory Peptides

The peptide sequence of the ultrafiltration fraction with high tyrosinase inhibitory activity was identified by Bio-Tech Pack Technology Co (Beijing, China). Nano LC-MS/MS was used to sequence the components. The experiment was carried out on the Easy-nLC 1200 liquid chromatograph and Q Exactive™ Hybrid Quadrupole-Orbitrap™ Mass Spectrometer (Thermo Fisher Scientific, Waltham, MA, USA). The original mass-spectrometry file was analyzed by Byonic software, and it was searched with the UniProt protein database according to the types of samples.

### 2.10. Molecular-Docking Studies

Ligands and protein required for molecular docking were prepared using AutoDock Vina software (http://vina.scripps.edu/, accessed on 3 July 2023). The crystal structures of the target proteins were downloaded from the PDB database (https://www.rcsb.org/structure/2Y9X, accessed on 22 June 2023) and then processed for removing water molecules, hydrogenating and modifying amino acids and adjusting force-field parameters [26]. The polypeptide structure was constructed by Discovery Studio 2020, and then energy-minimized. Finally, the receptor protein was docked to the peptide. The binding free energy (kcal/mol) of the target structure was representative of the binding capacity of the two, and the lower the binding free energy, the more stable the binding between the ligand and the receptor.

Pymol (https://pymol.org/2/, accessed on 12 July 2023) was used for visual analysis, and Discovery Studio 2020 Client (https://discover.3ds.com/discovery-studio-visualizer-download, accessed on 13 July 2023) was used for visual analysis of 2d images. We ran molecular docking and binding energy calculations with default parameters. The coordinates of the docking center between peptide and tyrosinase were (−9.13, −30.50, −45.86).

### 2.11. Synthesis of Tyrosinase Inhibitory Peptides

Depending on the dates of the LC–MS/MS analysis and virtual screening, four peptides with purity above 95% (*w*/*w*) were detected by HPLC: Tyr-Tyr-Pro (YYP), Phe-Arg-Val-Lys (FRVK), Pro-Tyr-Leu-Lys (PYLK), and Pro-His-His-Phe (PHHF). These were synthesized by the Aminolink Biotechnology Co., Ltd. (Shanghai, China). Mobile phase A, 0.1% trifluoroacetic in 100% water; mobile phase B, 0.1% trifluoroacetic in 100% acetonrtrile; flow rate, 1 mL/min; column, Inertsil ODS-SP (4.6 × 250 mm × 5 μm), with a detection wavelength of 220 nm. Eventually, the purified peptides were identified by ESI-MS spectroscopy.

### 2.12. Enzyme Inhibition Kinetic Assay

The enzyme concentration was plotted as the horizontal coordinate and the initial rate of the enzymatic reaction V (ΔOD _475/_min) was plotted as the vertical coordinate to determine whether it was reversibly inhibited or not. The Lineweaver–Burk double-inverse curve was obtained by taking the reciprocal of the substrate concentration (1/[S]) as the horizontal coordinate and the reciprocal of the initial reaction rate (1/V) as the vertical coordinate to determine the type of inhibition and to calculate the Mie constant (K_M_) and the maximum reaction rate (V_max_) [27].

### 2.13. Statistical Analysis

All data are presented as mean ± standard deviation (SD) of at least three different experiments. The response-surface data were analyzed by Design Expert 13. A multiple group comparison was conducted by one-way analysis of variance (ANOVA) and Duncan’s multiple-range test of IBM SPSS Statistics 23. A *p*-value of less than 0.05 was considered statistically significant.

## 3. Results

### 3.1. Protease Screening

The enzyme–substrate binding during enzymatic digestion is highly specific, and the protease species affects the tyrosinase inhibitory activity of the digest due to the different cleavage sites of different proteases [28]. It was discovered that the enzymatic products obtained from different enzymes had, in descending order, tyrosinase inhibitory activities of 81.18 ± 0.56% (papain), 75.48 ± 3.19% (neutral protease), 51.59 ± 0.69% (animal protease), 41.17 ± 2.70% (trypsin) and 16.42 ± 3.70% (alkaline protease) (Figure 1). Also, DH was closely related to tyrosinase inhibitory activity. The two enzymes with the highest DH—papain and neutral protease—produced enzymatic products with good tyrosinase inhibitory activity, with inhibition rates of 81.18 ± 0.56% and 75.48 ± 3.19%, respectively. Papain has a wider specificity than trypsin in that it cuts the carboxyl side of lysine and arginine residues as well as aromatic amino acid residues in the peptide chain, leaving the functional amino acid residues exposed on the terminal side of the peptide chain [29], so that the gelatinized peptide obtained has a higher inhibitory activity against tyrosinase. Papain was selected as the optimal enzyme based on the experimental results and the economic benefits.

### 3.2. Optimization of APGP

As shown in Figure 2A, both the tyrosinase inhibitory activity and the DH value of the enzymatic hydrolysate increased with the increase in enzyme dosage, and there was no significant difference in tyrosinase inhibitory activity when the enzyme dosage was greater than 5 U/mg. Considering the cost of the experiment, the optimum enzyme dosage of 5 U/mg was chosen. The pH values affect substrates and enzymes by altering the charge distribution of proteins, as well as by being able to cause changes in the exposure of the active site of enzymes [30]. The tyrosinase inhibitory activity and DH value of the enzymatic hydrolysate were significantly reduced in the range of pH 6.0–8.0, which was due to the high pH value affecting the binding of papain to the substrate, which led to the decrease in the enzyme digestion effect, so the optimal enzyme digestion pH value of 6.0 was selected (Figure 2B). During the continuous hydrolysis, the DH value increased continuously, and the tyrosinase inhibitory activity reached the maximum value at 4 h. Subsequently, the tyrosinase inhibitory peptide was gradually split into peptides or amino acids with smaller molecular weights, which led to a decrease in the tyrosinase inhibitory activity of the enzymatic hydrolysate, and thus, the optimal enzymatic hydrolysis time was selected to be 4 h (Figure 2C). At the stage of 45–55 °C, with the increase in temperature, molecular movement speed increased, enzyme activity rose and tyrosinase inhibitory activity and DH value increased; at high temperatures, the enzyme was prone to changes in its spatial structure and the peptide binding site with papain changed, reducing the enzymatic effect, so that at 65 °C and above, tyrosinase inhibitory activity decreased significantly (Figure 2D) (*p* < 0.05) [31].

### 3.3. Optimization Analysis of RSM

Further optimization of tyrosinase inhibitory activity was investigated using BBD-based RSM for three variables—enzymatic time (A), enzymatic temperature (B) and enzymatic pH (C)—in single-factor experiments. The experimental scheme and results are shown in Table 2. The experimental data in Table 3 were analyzed using Design Expert 13 software, and a multivariate regression model was established to obtain the quadratic multinomial regression equation between tyrosinase inhibition (*Y*) and enzymatic time (*A*), enzymatic temperature (*B*) and enzymatic pH (*C*) as follows:(3)$$Y=85.02-4.01A-2.71B-5.88C-9.95AB+4.07AC-4.97BC-28.31A^{2}-19.81B^{2}-28.63C^{2}$$
where *Y* is the tyrosinase inhibition (%).

Table 3 illustrates the model *p* < 0.0001, indicating that the difference between the models was extremely significant. The misfit term *p* = 0.5691 > 0.05, which is shown to be non-significant, indicates that the equation is well modeled and can be used well for data analysis. From the F value, the major and minor factors affecting the inhibitory activity of tyrosinase were C, A and B, in that order. The model fit coefficient R^2^ = 0.9945 and model corrected fit coefficient R^2^_adj_ = 0.9873 indicated that the model correlation and fit were high, and were able to accurately predict and analyze the effects of the variables on tyrosinase inhibition [32].

Three-dimensional response-surface maps were generated by quadratic equations, with the value of the independent variable at the center point kept constant. The eccentricity of the ellipse in the contour plot in the lower part of the image reflects the degree of influence of the independent variable on tyrosinase inhibitory activity [33]. The three-dimensional graphs are depicted in Figure 3, the centroids of the response surfaces of the three-dimensional graphs were all within the range of the variables, and the contour plots indicate that there was a strong interaction between any two factors, with the degree of influence in descending order of magnitude: A–C, B–C and A–B.

Response-surface analysis demonstrated that the optimal enzymatic hydrolysate conditions were an enzyme temperature of 54.6 °C, pH of 5.9 and time of 3.96 h, resulting in a tyrosinase inhibition of 85.5%. And the tyrosinase-inhibiting activity of the validation experiment was 88.6% (modified according to actual experimental conditions: 54.5 °C, 238 min, pH 5.8), which was less different from the theoretical prediction, and the optimization result of the model was considered to be more accurate.

### 3.4. Amino Acid Composition and MWD of APGP

Table 4 shows the amino acid composition of the APGP; 16 amino acids were detected, and the total amount of amino acid was 69.3 ± 0.29 g/100 g. Effective tyrosinase-inhibiting peptides typically contain hydrophobic residues and basic amino acid residues [34]. Aliphatic hydrophobic amino acid residues of Ala, Val, Leu, Ile and Met can directly interact with the enzyme to inhibit dopaquinone formation, and thus, the production of melanin [35], and these five amino acids accountd for 20.86% of the total amino acid content. Uncharged polar amino acids such as Ser, Cys and Thr residues, which may inhibit melanin formation by conjugating with the semiquinone group, an intermediate product in the melanogenesis process [22], had a total content of 6.40 g/100 g. Tyr residues can activate tyrosinase and form a competitive inhibition with it, competing with tyrosinase substrates at the active site of tyrosinase [36]. His modification can directly expose the hydrophobic group and disrupt the conformation of tyrosinase, thus affecting the activity of tyrosinase itself. And His and Arg residues can interact with Cu ions in the active center of tyrosinase to inhibit tyrosinase activity [37,38]. The contents of these three amino acids in the AGPG were 1.66 g/100 g, 0.64 g/100 g and 6.19 g/100 g, respectively, accounting for 16.45% of the total amino acid content. It was found that good tyrosinase binding and inhibitory peptides are usually combinations of Arg and or Phe with Val, Ala and/or Leu [39]. The percentage of the above functional amino acids in the amino acid composition of the APGP was as high as 42.33%, indicating that APGP is promising for the study of tyrosinase inhibitory activity.

The retention times (Rts) of the PEG standards of different MWs were in the following order: 17.3 min (326 kDa), 18 min (152 kDa), 19 min (78.3 kDa), 19.8 min (44 kDa), 20.7 min (25.3 kDa), 23.4 min (4.29 kDa), 25.3 min (1.4 kDa) and 28.0 min (0.43 kDa). The equation log(MW) = 10.01334 − 0.26857t was obtained with Rt as the horizontal coordinate and log(MW) of the molecular mass of the standard as the vertical coordinate, which was linear with linear correlation coefficient R^2^ = 0.99019. The MWD of peptides reflects the degree of hydrolysis [40]. As shown in Table 5 and Figure 4, the MWD of APGP ranged from 18,994 to 5010 Da (relative content 16.74%); 4989 to 3010 Da (relative content 16.91%); 2998 to 1001 Da (relative content 33.50%) and 998 to 314 Da (relative content 15.57%). In summary, the MW of APGP was mainly concentrated between 1 and 5 kDa, so we then used 5 kDa and 3 kDa ultrafiltration membranes for the next step of separation.

### 3.5. Enrichment and Identification of Potential APGPs

In general, lower MW enzymatic peptides have higher potential biological activity. Inhibition of tyrosinase by gelatin peptides of different MWs is similarly influenced by amino acid composition and peptide sequence [41]. In order to compare the tyrosinase inhibitory activities of gelatin peptides with different MWs, in this study, we chose APGPs with higher tyrosinase inhibitory activities for the next step of ultrafiltration separation, and we obtained the three ultrafiltration fractions, namely, APGP-I (>5 kDa), APGP-II (3–5 kDa) and APGP-III (<3 kDa), and the tyrosinase inhibitory activities of the fractions are shown in Table 6. APGP-III showed the strongest inhibitory effect on tyrosinase (IC_50_ of 0.608 ± 0.021 mg/mL), whereas APGP-I showed the least inhibitory activity on tyrosinase (IC_50_ of 6.863 ± 0.098 mg/mL), with a more pronounced molecular-weight-dependence, which was similar to that of the ultrafiltration of separated fractions from the fish skin gelatin peptides [17].

In order to identify peptides with tyrosinase inhibitory activity, the sequence peptides of APGP-III were characterized by LC-MS/MS. Peptide sequence resolution of the mass spectrometry raw files was performed using PEAKS Studio (8.5) software, and a total of 200 peptides were identified. Water solubility and toxicity are usually important indexes for evaluating the metabolic transit distribution of drugs in the body, and the prerequisite for bioactive peptides to exert their biological activities in the organism is, first and foremost, the absence of toxic effects on the organism [42]. Peptides with MW < 1 kDa and score > 100 were searched through the website (https://webs.iiitd.edu.in/raghava/toxinpred/index.html, accessed on 15 July 2023). Eight non-toxic peptides with GRAVY value < 0 were screened out, and then the data were searched on the website of http://www.uwm.edu.pl/, accessed on 15 July 2023. It was found that these eight peptides were unreported, and these could be used to further investigate the molecular mechanism of peptide inhibition of tyrosinase.

### 3.6. Mechanism of Tyrosinase Inhibition

#### 3.6.1. Molecular Docking

Molecular docking is a common method of simulating the way molecules interact with each other and predicting their binding modes and affinities through a computerized platform [43]. This method is widely used in polypeptide activity, where the lower the intermolecular binding energy suggests a better peptide−protein affinity, and theoretically a stronger tyrosinase inhibitory activity of the peptide segment [44]. The processed eight peptides and kojic acid as a small molecule ligand and tyrosinase as a protein receptor were molecularly docked by AutoDock Vina. The docking binding-energy scores ranged from −5.4 to −7.6 kcal/mol (Table 7). The predicted binding energies of YYP, PHHF, FRVK and PYLK were −7.6 kcal/mol, −7.3 kcal/mol, −7.0 kcal/mol and −6.9 kcal/mol, respectively, indicating that these four peptides have theoretical tyrosinase inhibitory activity. In terms of amino acid composition, PHHF and FRVK have the hydrophobic aromatic amino acid Phe, and His and Arg residues, which could interact with the active-site Cu ion [38]. YYP and PYLK are present to activate the Tyr residue of the enzyme, and the hydrophobic aromatic amino acid Leu. The screening results were consistent with the amino acid analysis (Table 4). Similar naturally occurring peptides have been identified in other studies, such as DLGFLARGF isolated from fish scale gelatin peptides, with the presence of four amino acid residues, Leu, Phe, Ala and Arg, which are tightly linked to tyrosinase inhibitory activity [17].

Protein−ligand interactions are essential in molecular-docking studies. In order to more visually show how peptides bind to tyrosinase, a visual analysis was performed using the Discovery Studio 2020 Client tool. The crystal structure of *Agaricus bisporus* tyrosinase (2Y9X) is in the form of a homotetramer, in which the active site consists of two copper atoms and six histidine residues in coordination (His61, His85, His94, His259, His292 and His296) [8]. Peptide PHHF and peptide FRVK have the amino acid residue Phe, which is structurally similar to tyrosine and could be used as a substrate to bind to the active site of tyrosinase and, in this way, show good affinity [35]. Both peptide PYLK and peptide YYP interact with His85 at the active site to form hydrogen bonds and enhance tyrosinase stability. Peptide PYLK forms two hydrogen bonds with amino acid residues Gly281 and His85 around the tyrosinase A-chain, while peptide YYP interacts with amino acid residues around the A-chain of tyrosinase, forming six hydrogen bonds on the A-chain with Met280, Gly281, Asn260, Arg268, His244 and His85 (Figure 5). The formation of hydrogen bonds helps to increase the hydrophobicity of the receptor protein, which plays an important role in the strength and stability of the complex binding [45]. The results showed that peptide YYP had a better binding energy of −7.6 kal/mol, compared to peptide PYLK, with a binding energy of −6.9 kcal/mol. In addition, amino acid residues such as His85, Val283 and Phe264 can form a hydrophobic pocket on the enzyme A-chain, which wraps the peptide YYP tightly by hydrophobic interaction. Also, electrostatic interactions occurred between Glu256 and the peptide PPY on the A-chain. These results intuitively indicate that hydrogen bonding is the main driving force involved in the interactions (Table 8).

#### 3.6.2. Synthetic Peptide Activity Validation

Therefore, four peptides with the lowest binding energies were synthesized and validated for in vitro activity (Table 9). The trends in tyrosinase inhibitory capacity of the four peptides were not entirely consistent with the results of the binding-energy scores. As shown in Table 8, peptides PHHF and PYLK had lower tyrosinase inhibitory activity, presumably related to their lower driving force. Peptide YYP had the highest tyrosinase inhibitory activity (IC_50_ of 1.764 ± 0.025 mM), followed by peptide FRVK (IC_50_ of 5.873 ± 0.068 mM), and both peptides showed good tyrosinase inhibitory activity. Compared with other natural plant extracts, such as *Dalbergia ecastaphyllum* and *Prunus persica* [46,47], peptide FRVK and peptide YYP had relatively similar tyrosinase inhibitory abilities, however, the inhibitory activity of peptide FRVK still fell short of valerian tannins (*Quercus acutissima Carr*) and Apios isoflavone glucoside (*Apios americana*) [48,49]. Therefore, we chose the more active peptide YYP for subsequent analytical experiments.

#### 3.6.3. Detection of Inhibitory Effect, Mechanism and Type

To elucidate the mechanism of inhibition of tyrosinase by YYP, kinetic studies were used to investigate the kinetic characterization of the inhibitory response (Figure 6). The enzyme concentration ([E]) was plotted against the initial reaction rate ([V]) at different YYP concentrations, and the results manifested a set of straight lines passing through the origin, and all the lines showed a good linear relationship. As the concentration of YYP increased, the slope of the straight line gradually decreased, indicating that the addition of YYP did not change the amount of enzyme, but decreased the ability to decompose the substrate by reducing the activity of the enzyme, so the inhibition of YYP on the enzyme was reversible, and the results of the present experiments were more consistent with those of previous research [50,51].

The Lineweaver–Burk double-inverse plot of the inhibitory effect of YYP on tyrosinase is illustrated in Figure 6. The 1/V were both fitted linearly to the 1/[S] (R^2^ and p-values shown in Table 10) and intersected in the second quadrant of the graph. As the amount of YYP increased, the value of K_M_ increased and the value of V_max_ decreased. Inhibitory classes were consistent with a mixture of noncompetitive and competitive types.

The mixed-inhibitory type inhibits tyrosinase via two distinct pathways: competitively forming enzyme–inhibitor (EI) complexes and interrupting enzyme–substrate–inhibitor (ESI) complexes in a noncompetitive manner. The inhibition constant K_I_ or K_IS_ is the dissociation constant for the dissociation of the inhibitor (I) from the EI complexs or the enzyme–inhibitor–substrate (EIS) complexs. Both baicalein and cinnamic acid derivatives are reversible mixed-type inhibitors [52,53]. Methoxy-substituted tyramine derivatives acted as hybrid inhibitors with stronger binding between the enzyme and the compounds, indicating that the enzyme prefers a competitive approach (K_I_ < K_IS_) [54]. Summarized in Table 10, based on the slope ratio and intercept ratio of the Lineweaver–Burk curve (with and without the presence of samples), the K_I_ and K_IS_ can be obtained [51]. K_I_ and K_IS_ gradually increased with increasing concentration of YYP, and 1/K_IS_ > 1/K_I_, indicating that the affinity of YYP for tyrosinase was greater than that of the complex of L-tyr and tyrosinase. Furthermore, the inhibitory effect of YYP was manifested by entering the active center of tyrosinase or binding to the surface of tyrosinase, confirming that the type of inhibition of tyrosinase by YYP was a mixed type of inhibition, consistent with the above results.

## 4. Conclusions

In the present study, we found that the gelatin peptide from *Atrina pectinata* mantle has good tyrosinase inhibition potential. The tyrosinase inhibitory active peptide was optimally prepared using one-way and response-surface experiments, and the inhibition rate was 88.6% (IC_50_ of 3.268 ± 0.048 mg/mL). APGP was isolated and purified by ultrafiltration to obtain the highly inhibitory active fraction APGP-III (IC_50_ of 0.608 ± 0.021 mg/mL). The inhibitory peptide YYP was identified from APGP-III for the first time by LC–MS/MS and virtual screening. Inhibition kinetics and molecular-docking results showed that YYP is a reversible hybrid inhibitor with an IC_50_ value of 1.764 ± 0.025 mM, the main driving force for interaction with tyrosinase is hydrogen bonding, and YYP has a good binding affinity with a binding energy of −7.60 kcal/mol. From our results, it can be concluded that the peptide YYP obtained from *Atrina pectinata* mantle is efficient and can be used for the subsequent development of new and effective melanogenesis inhibitors.

## Figures and Tables

**Figure 1 foods-12-03884-f001:**
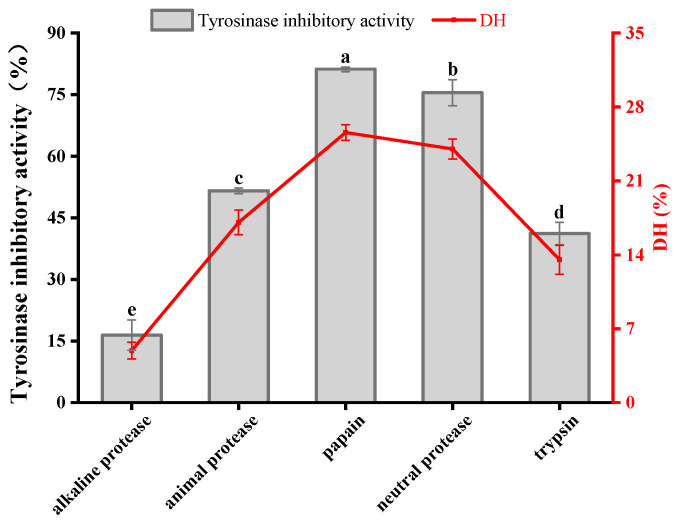
Effect of five proteases on tyrosinase inhibition and DH during hydrolysis. Different superscript letters in the same group indicate that they are significantly different (*p* < 0.05).

**Figure 2 foods-12-03884-f002:**
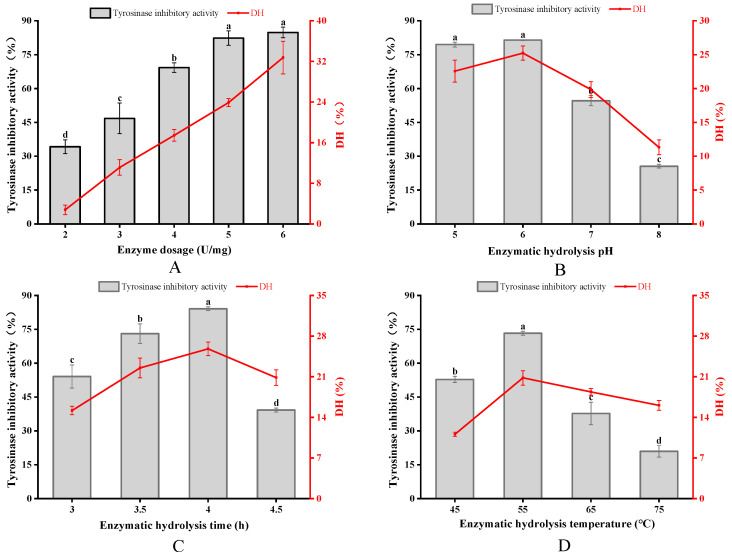
Effects of different enzymatic hydrolysis factors on tyrosinase inhibitory activity and DH value. Enzyme dosage (**A**), pH (**B**), time (**C**) and temperature (**D**). Different superscript letters indicate that they are significantly different (*p < 0.05*).

**Figure 3 foods-12-03884-f003:**
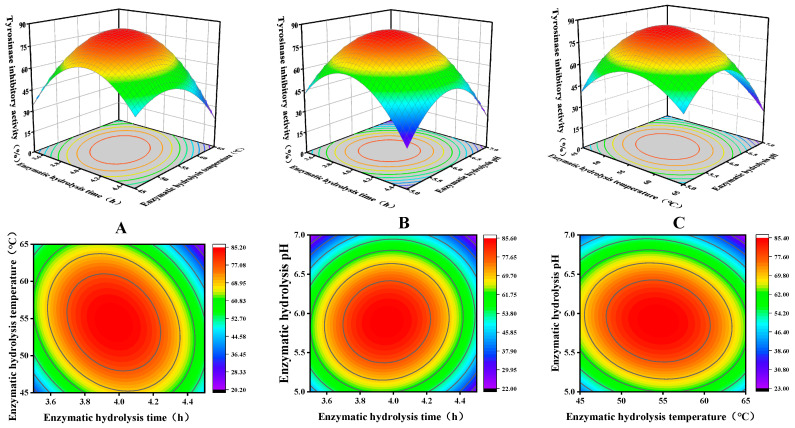
Response surface showing the impact of interactions between independent variables on the activity of tyrosinase inhibition. (**A**) The interaction between time and temperature, (**B**) the interaction between time and pH and (**C**) the interaction between temperature and pH.

**Figure 4 foods-12-03884-f004:**
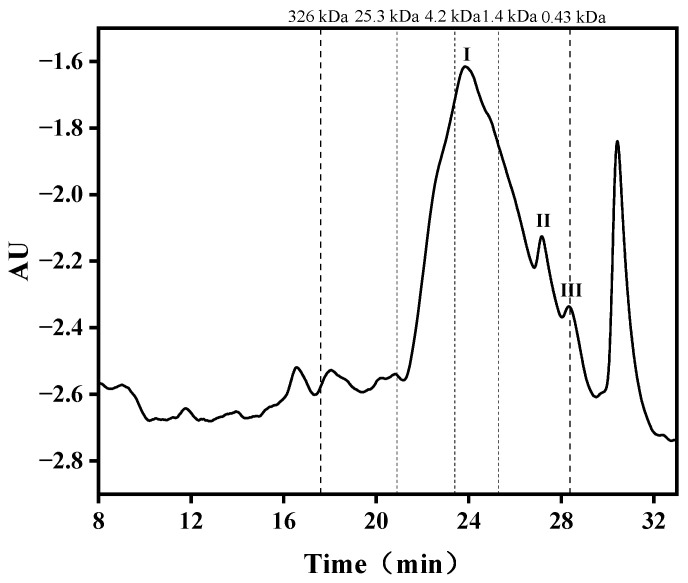
The gel permeation chromatogram of APGP-I, APGP-II and APGP-III represents the different components obtained.

**Figure 5 foods-12-03884-f005:**
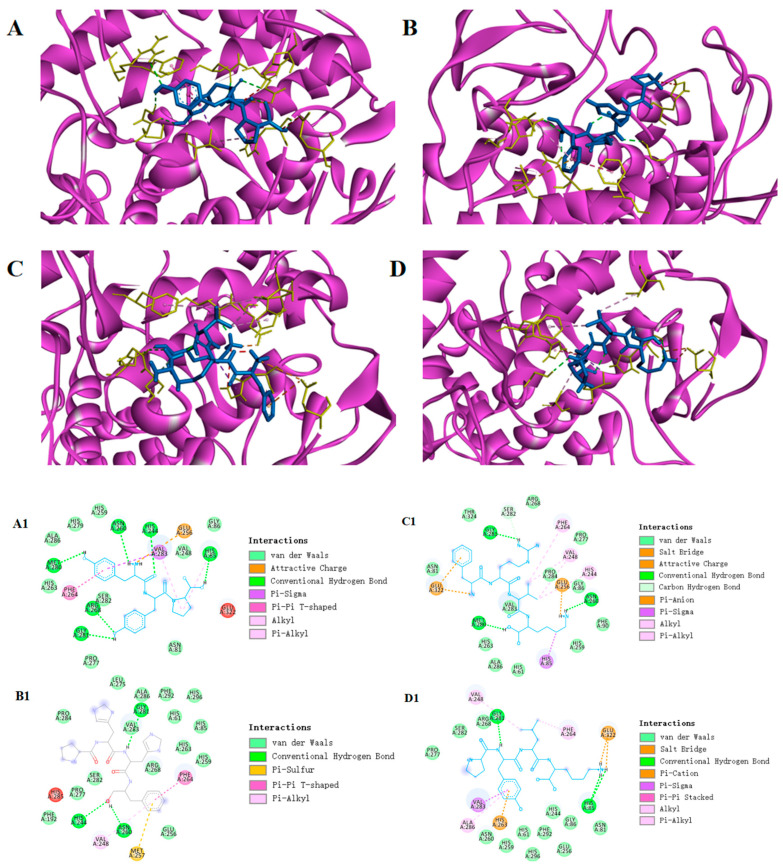
Hydrophobic interaction between YYP, PHHF, FRVK and PYLK and amino acid residues of tyrosinase (**A**–**D**,**A1–D1**).

**Figure 6 foods-12-03884-f006:**
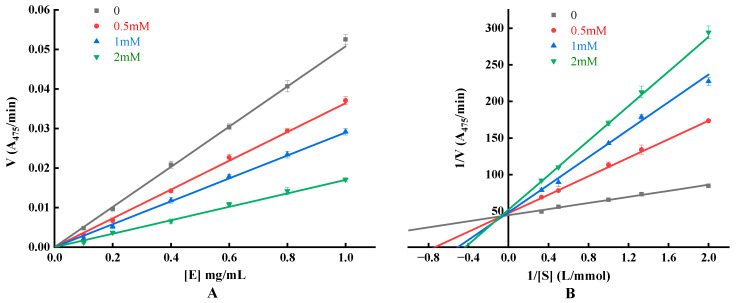
Graph of enzyme concentration and initial reaction rate (**A**) and Lineweaver–Burk plots of YYP (**B**).

**Table 1 foods-12-03884-t001:** Table of tyrosinase inhibitory activity reagents.

Reagent	Solvent Base Well T_1_, (μL)	Solvent Reaction Well T_2_, (μL)	Sample Background Well T_3_, (μL)	Sample Reaction Well T_4_, (μL)
L-Tyr solutions	0	40	0	40
Sample solutions	0	0	40	40
PBS	110	70	70	30
Mushroom tyrosinase solution	20	20	20	20

**Table 2 foods-12-03884-t002:** Design and results of response-surface experiment.

Number	A (Time/h)	B (Temperature/°C)	C (pH)	Inhibition (%)
1	0	0	0	83.9
2	−1	0	−1	41.3
3	0	0	0	85.1
4	0	1	−1	45.6
5	0	0	0	82.3
6	0	0	0	89.7
7	1	0	1	23.0
8	0	−1	1	37.5
9	0	1	1	21.1
10	−1	−1	0	32.5
11	1	1	0	21.4
12	−1	0	1	24.2
13	0	0	0	85.1
14	0	−1	−1	42.1
15	1	−1	0	45.7
16	1	0	−1	23.8
17	−1	1	0	48.0

**Table 3 foods-12-03884-t003:** Regression model and ANOVA results based on tyrosinase inhibitory activity.

Source	Sum of Squares	df	Mean Square	F Value	*p*-Value	Significance
Model	10,466.84	9	1162.98	139.63	<0.0001	**
A-time	128.80	1	128.80	15.46	0.0057	**
B-temperature	58.86	1	58.86	7.07	0.0325	*
C-pH	276.13	1	276.13	33.15	0.0007	**
AB	396.01	1	396.01	47.55	0.0002	**
AC	66.42	1	66.42	7.89	0.0256	*
BC	99.00	1	99.00	11.89	0.0107	*
A^2^	3374.55	1	3374.55	405.17	<0.0001	**
B^2^	162.36	1	162.36	198.40	<0.0001	**
C^2^	3452.48	1	3452.48	414.53	<0.0001	**
Residual	58.30	7	8.33			
Lack of fit	21.29	3	7.10	0.7671	0.5691	
Pure error	37.01	4	9.25			
Cor total	10,525.14	16				

Significance: ** very significant, *p* < 0.01; * significant, *p* < 0.05.

**Table 4 foods-12-03884-t004:** Amino acid composition of APGP.

Amino Acid	Content (g per 100 g)	Amino Acid Ratio (%)	Amino Acid	Content (g per 100 g)	Amino Acid Ratio (%)
Glu	11.3 ± 0.14	16.32 ± 0.27	Thr	3.08 ± 0.02	4.44 ± 0.01
Gly	11.7 ± 0.14	16.89 ± 0.27	Val	2.94 ± 0.07	4.25 ± 0.12
Asp	7.04 ± 0.07	10.17 ± 0.14	Ser	3.32 ± 0.01	4.80 ± 0.03
Arg	6.19 ± 0.18	8.94 ± 0.23	Ile	1.98 ± 0.13	2.87 ± 0.18
Leu	3.52 ± 0.08	5.08 ± 0.10	Tyr	1.66 ± 0.10	2.40 ± 0.13
Pro	5.38 ± 0.01	7.78 ± 0.02	Phe	1.44 ± 0.06	2.07 ± 0.08
Ala	4.22 ± 0.01	6.09 ± 0.02	Met	1.78 ± 0.06	2.57 ± 0.07
Lys	3.07 ± 0.03	4.43 ± 0.02	His	0.64 ± 0.04	0.92 ± 0.05

**Table 5 foods-12-03884-t005:** MWD of APGP.

MW Range (Da)	Rt (min)	Relative Content (%)
18,994–5010	21.132–23.198	16.74
4989–3010	23.206–24.091	16.91
2998–1001	24.098–26.302	33.50
998–314	26.309–29.304	15.57

**Table 6 foods-12-03884-t006:** Tyrosinase inhibitory activity (IC_50_ mg/mL) of the unfractionated APGP and the fractions obtained via ultrafiltration.

Component	Tyrosinase Inhibitory Activity (IC_50_, mg/mL)
APGP	3.268 ± 0.048 ^c^
APGP-I (MW > 5 kDa)	6.863 ± 0.098 ^d^
APGP-II (3 kDa < MW < 5 kDa)	1.372 ± 0.039 ^b^
APGP-III (MW < 3 kDa)	0.608 ± 0.021 ^a^

All values are expressed as mean tyrosinase inhibitory activity (IC_50_ values) ± standard deviation and tests were performed in triplicate. The presence of different superscript letters in the same column indicates a significant difference (*p* < 0.05).

**Table 7 foods-12-03884-t007:** Potential tyrosinase inhibitory peptide sequences: screening by mass spectrometry and in silico.

Peptide Sequence	Protein Name	Intensity	*m/z*	Score	Toxin	Gravy	XP Score (kcal/mol)
THG	A0A7R7DZB7	56,383,000	314.1472	150.1	Non-toxin	−1.43	−5.4
LGLKNK	L0ET25	21,389,000	336.7240	168.09	Non-toxin	−0.68	−6.3
PYNVVHT	A0A7R7DZL1	189,410,000	829.4233	137.91	Non-toxin	−0.27	−6.5
VPKLH	A0A7R7DZL1	10,988,000	297.1922	252.12	Non-toxin	−0.14	−6.8
PYLK	A0A7R7DZB4	30,099,000	260.6599	128.3	Non-toxin	−0.75	−6.9
FRVK	A0A7R7I1R9	23,149,000	275.1791	231.86	Non-toxin	−0.35	−7.0
PHHF	L0ER83	49,520,000	269.132	286.64	Non-toxin	−1.30	−7.3
YYP	L0ET20	17,004,000	442.1976	133.49	Non-toxin	−1.40	−7.6

**Table 8 foods-12-03884-t008:** Docking results according to hydrophobic interaction, electrostatic interaction and hydrogen bonds.

	PHHF	PYLK	FRVK	YYP
Hydrogen bonds	His244, Asn260, Gly281	Gly281, His85	Met280, Gly281, Asn260, Ser282	Met280, Gly281, Asn260, Arg268, His244, His85
Electrostatic interaction	NO	Glu322, His263	Glu322, Glu256	Glu256
Hydrophobic interaction	Val248, Phe264	Phe264, Val248, His263, Val283, Ala286,	Phe264, Val248, His244, His85	His85, Phe264, Val283

**Table 9 foods-12-03884-t009:** Tyrosinase inhibitory activity of synthetic peptides.

Peptide Sequence	Tyrosinase Inhibitory Activity (IC_50_, mM)
YYP	1.764 ± 0.025
FRVK	5.873 ± 0.068
PHHF	>10
PYLK	>10

**Table 10 foods-12-03884-t010:** Inhibitory kinetic parameters of YYP on tyrosinase.

Concentration (mM)	Fit Equation	Coefficient of Determination R^2^	Michaelis’s Constant K_M_ (mmol/L)	Maximum Reaction Speed V_max_ (A_475_/min)	Inhibition Constant K_I_ (mM)	Inhibition Constant K_IS_ (mM)
0	y = 20.7837x + 44.6125	0.9839	0.4659	0.0224	-	-
0.5	y = 63.1532x + 47.2733	0.9992	1.3359	0.0212	0.2453	8.3833
1	y = 94.1308x + 48.5211	0.9928	1.9400	0.0206	0.2834	11.4139
2	y = 118.0196x + 51.9697	0.9968	2.2709	0.0192	0.4275	12.1276

## Data Availability

The data showed in this study are contained within the article.

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
