# Peer review of "Optimization of a Novel Tyrosinase Inhibitory Peptide from *Atrina pectinata* Mantle and Its Molecular Inhibitory Mechanism"

_foods, 2023, doi:10.3390/foods12213884_

Round 1
Reviewer 1 Report
Comments and Suggestions for Authors
The by-products of Atrina pectinate was explored, which gelatin was extracted from the mantle and the potential whitening effect of its enzymatic peptides were explored. Taking tyrosinase inhibitory activity as the evaluation index, the enzyme hydrolysate process was optimized by response surface methodology, and the optimal enzyme hydrolysate conditions were as follows: pH 5.82, 238 min enzyme hydrolysate time, and temperature of 54.5 ℃. Under these conditions, the tyrosinase inhibition of APGP was 88.6% (IC50 of 3.268±0.048 mg/mL). The peptides obtained from the identification were separated by ultrafiltration and LC-MS/MS, and then four new peptides were screened by molecular docking, among which the peptide Tyr-Tyr-Pro (YYP) had the strongest inhibitory effect on tyrosinase with an IC50 value of 1.764±0.025 mM. Lineweaver-Burk analysis and molecular docking indicate.
The research is good design and well organized. However, there still have some issues need to check.
1. Please confirm the expression in Materials.
2. The activity of peptide should evaluate based on its structure. Please refer this reference (Journal of Agricultural and Food Chemistry, 2012, 60(45):11351-11358).
3. The isolated chromatography should be supplement.
4. 3.6.1. Molecular Docking. The mechanism about QSAR should be introduce based on the molecular docking about: Hydrogen bonds, electrostatic interaction, Hydrophobic interaction.
5. The data should be expressed with standard error for M±SD.
6. It is better to supplement HPLC spectrogram.
7. The expression should be verified in the manuscript.
8. The reference should be updated in recent years.
Comments on the Quality of English Language
The by-products of Atrina pectinate was explored, which gelatin was extracted from the mantle and the potential whitening effect of its enzymatic peptides were explored. Taking tyrosinase inhibitory activity as the evaluation index, the enzyme hydrolysate process was optimized by response surface methodology, and the optimal enzyme hydrolysate conditions were as follows: pH 5.82, 238 min enzyme hydrolysate time, and temperature of 54.5 ℃. Under these conditions, the tyrosinase inhibition of APGP was 88.6% (IC50 of 3.268±0.048 mg/mL). The peptides obtained from the identification were separated by ultrafiltration and LC-MS/MS, and then four new peptides were screened by molecular docking, among which the peptide Tyr-Tyr-Pro (YYP) had the strongest inhibitory effect on tyrosinase with an IC50 value of 1.764±0.025 mM. Lineweaver-Burk analysis and molecular docking indicate.
The research is good design and well organized. However, there still have some issues need to check.
1. Please confirm the expression in Materials.
2. The activity of peptide should evaluate based on its structure. Please refer this reference (Journal of Agricultural and Food Chemistry, 2012, 60(45):11351-11358).
3. The isolated chromatography should be supplement.
4. 3.6.1. Molecular Docking. The mechanism about QSAR should be introduce based on the molecular docking about: Hydrogen bonds, electrostatic interaction, Hydrophobic interaction.
5. The data should be expressed with standard error for M±SD.
6. It is better to supplement HPLC spectrogram.
7. The expression should be verified in the manuscript.
8. The reference should be updated in recent years.
Author Response
Dear Reviewer,
We have uploaded the "Response to Reviewer Comments" for the paper. Please see the attachment.
Kind regards,
Wen Wang
E-Mail:wangwen31@stu.gdou.edu.cn
Response to Reviewer Comments
Thanks for the comments from Reviewer. First of all, we’d like to stress that we totally respect the hard work the reviewer and editor made to improve our manuscript. We are much appreciated to have this chance to set the problem straight.
The main responses are showed as following:
#Responses: Reviewer:
- Please confirm the expression in Materials.
Response 1: We have added a description of the source of the gelatin as follows : “Atrina pectinata was purchased from the local Dongfeng Market in Zhanjiang, China, and the methods of gelatin from mantle (APG) preparation referenced to wang et al [22].” In addition, lowercase the initial letters of the proteases in the ingredients, e.g. “papain”, “animal protease” and “neutral protease”. Moreover, we've changed “12,000 r/min” to “12,000 rpm/min” and added the full name , e.g. “Response Surface Methodology (RSM)”, “phosphate buffer solution (PBS, pH 6.8, 0.1 mol/L)” to the revised manuscript. We've made additional description in the Method section “2.2” and please see the red-lettered section of the revised draft for changes elsewhere.
- The activity of peptide should evaluate based on its structure. Please refer this reference (Journal of Agricultural and Food Chemistry, 2012, 60(45):11351-11358).
Response 2: Thanks for Reviewer’s kind comments. We have carefully read this reference you provided and realized the error, which has been corrected as “Effective tyrosinase-inhibiting peptides typically contain hydrophobic residues and basic amino acid residues [34] ”.
34.Liu, L.L.; Liu, L.Y.; Lu, B.Y.; Xia, D.Z.; Zhang, Y. Evaluation of antihypertensive and antihyperlipidemic effects of bamboo shoot angiotensin converting enzyme inhibitory peptide in vivo. J. Agr. Food Chem. 2012, 60, 11351-11358.
- The isolated chromatography should be supplement.
Response 3: Regarding APGP, we determined its molecular weight distribution (the gel permeation chromatogram of APGP was shown in Figure 4) and then directly took the ultrafiltered fraction (APGP-III) of <3 KD for LC-MS/MS analysis without further isolation and purification.
- The 3.6.1. Molecular Docking. The mechanism about QSAR should be introduce based on the molecular docking about: Hydrogen bonds, electrostatic interaction, Hydrophobic interaction.
Response 4: The binding affinity between proteins and ligands is affected by various types of interactions. These interactions include hydrogen bonding, hydrophobic interactions, electrostatic interactions, etc. We have added a description of these interactions in 3.6.1 as follows: “Protein-ligand interactions are essential in molecular docking studies. Electrostatic interactions, including ionic and salt bridges (resulting from attraction/repulsion between charged groups, contribute to the overall binding affinity; the Electrostatic interactions, including ionic and salt bridges (resulting from attraction/repulsion between charged groups), contribute to the overall binding affinity; hydrophobic interactions play an important role. Hydrophobic interactions play an important role in stabilizing binding between hydrophobic ligands and hydrophobic binding pockets on proteins; and hydrogen bonding plays an important role in determining the specificity and strength of binding.”
- The data should be expressed with standard error for M±
Response 5: We have checked the manuscript and corrected all data that appeared to be irregularly expressed, such as “inhibition rates of 81.18±0.56 % and 75.48±3.19 %” on page 6 and 3.6.2 “tyrosinase inhibitory activity (IC50 of 1.764±0.025 mM), followed by peptide FRVK (IC50 of 5.873±0.068 mM)” on page 13. The revised details could be found in the new revision (3.1, 3.4, 3.6.2, revision).
- It is better to supplement HPLC spectrogram.
Response 6: Thanks for this helpful comment. We supplemented the elution chromatograms of APGP and labeled the graph with the peak times of the molecular standardsin 3.4 (Figure 4 , revision).
- The expression should be verified in the manuscript.
Response 7: We have done our best to completely revise our manuscript based on all of the above comments. We have added a description of the raw material Atrina pectinata and its mantle in the introductory section (Page 2, revision), and the scientific name of the plants have been added and italicized in the 3.6.2 analysis section. The revised details could be found in the new revision.
- The reference should be updated in recent years.
Response 8: Thanks very much for the good comment and your hard work. We have updated six new references in recent years, with the number of references [2, 3, 5, 7, 9, 12]. The notes in the text have been modified accordingly and marked in red.
We have tried our best to revise our manuscript overall according to all the comments above. On behalf of my co-authors, we thanks again for giving us so much kind comments and suggestions.
Reviewer 2 Report
Comments and Suggestions for Authors
Foods-2624397-peer-review-v1
Major comments:
This is a study that estimated the peptide sequence with tyrosinase inhibitory effect by molecular docking, actually synthesized the peptide to verify the effect, and confirmed that the YYP oligopeptide has tyrosinase inhibitory effect. The topics of gelatin degradation products and amino acid composition obtained from the mantle of shellfish seem to deviate from the main topic, and if the focus is on YYP oligopeptides, the paper would be cleaner if the results extracted from shellfish were removed. Alternatively, if the authors want to discuss oligopeptides obtained from shellfish, they need to indicate what oligopeptides were contained in the fraction and the ratio of each peptide present.
There were many sentences that did not include necessary information or were oversimplified. It would be better to resubmit the paper after it has been peer-reviewed by a supervisor.
The author's affiliation #1 seems curious. How many laboratories are there?
P1L7, in abstract section: I think "the tyrosinase inhibition activity" more better. What is APGP? Please define the term.
Lineweaver-Burk analysis often reveals the maximum activity and half-maximum values but it could not realized the molecular-molecular interaction force. Please rephrase the last 4 lines.
Introduction section
L2: Use ultraviolet here before page2-line1.
Second paragraph line4, “tric acid” means “citric acid”?
3rd paragraph line4, It’s preferred that “ACE” first define the term then abbreviate as ACE.
Line8-9. Did Gelatin peptide from fish scales show 61.7% inhibition or 61.7% inhibitory activity? What is 100% of tyrosinase inhibitory activity?
∆MI probably melanin index but it is preferred full spell in first.
The last paragraph is the first mention of Atrina pectinata, but there is no description of why the focus was on the mantle of Atrina pectinata. I would like to see a more detailed description of why Atrina pectinata and why the focus was on the Atrina pectinata mantle.Please explain why the tyrosinase inhibitory peptide was known to exist in the Atrina pectinata mantle in the first place.
Finally, is it correct to conclude that the author is not talking about a peptide obtained from Atrina pectinata, but an oligopeptide sequence YYP that has tyrosinase inhibitory activity?
In Materials and methods section:
2.1. Line1, the laboratory is which one? If “the laboratory” prepared the gelatin previously, author should state the references. Probably, “tyrosinase.”
2.2, What is enzymolysis? How many and how much protease were used at which pH and what temperature? Too little information on proteolytic conditions.
12000 r/min => Show as a gravitational acceleration, or at least rpm.
2.3, What is RSM?
2.4 Line5, What is 0.1 M PBS? Where did the “As” go?
Is equation of (1) correct? Do A4 and A3 not subtract A1? What is mean “hole”? Did you analyzed these absorbance using 96-well plate?
Table 1, The column Reagent/µL, separate Reagents and solution volume unit. The volume unit put the column solvents (T1-T4).
2.7, 2.8, If author want to use abbreviation of MW, use the MW instead of molecular weight there after.
2.7, Line6, missing parentheses “PEG”.
2.8, Line1, What is MWCO? State correctly, probably molecular weight cut off.
P9, 3.4, Line13-15 and Table4, This is my suggestion. Since Table 4 shows amino acid content, how about adding amino acid ratio information for each amino acid in Table 4, as in line 15?
Is the term “enzymatic peptide” means enzymatic-digested peptide? Or a peptide has enzymatic activity?
P10, line2-12 and Table5, Why don’t author display these textual information as as figures (plot graph)? I would think that would give clearer and more informative results.
Moreover, I would like to see the elution profile of APGP along with the molecular standard profile.
Table6 title indicated unfractionated APGP but table 6 seems only fractionated APGP data. The data missing? In footnote, “Std. Deviation” should be “standard deviation”.
P11, 3.6.1 regarding molecular docking section, Are the 10 peptides synthetic or textual sequences? If those peptides were synthesized, table 7 was in vitro results. But textual sequences, the results mean in silico?
P13, Scientific names of plants should be italicized.
Author Response
Dear Reviewer,
We have uploaded the "Response to Reviewer Comments" for the paper.
Please see the attachment.
Kind regards,
Wen Wang
E-Mail:wangwen31@stu.gdou.edu.cn
Response to Reviewer Comments
Thanks for the comments from Reviewer. First of all, we’d like to stress that we totally respect the hard work the reviewer and editor made to improve our manuscript. We are much appreciated to have this chance to set the problem straight.
The main responses are showed as following:
Major comments:
This is a study that estimated the peptide sequence with tyrosinase inhibitory effect by molecular docking, actually synthesized the peptide to verify the effect, and confirmed that the YYP oligopeptide has tyrosinase inhibitory effect. The topics of gelatin degradation products and amino acid composition obtained from the mantle of shellfish seem to deviate from the main topic, and if the focus is on YYP oligopeptides, the paper would be cleaner if the results extracted from shellfish were removed. Alternatively, if the authors want to discuss oligopeptides obtained from shellfish, they need to indicate what oligopeptides were contained in the fraction and the ratio of each peptide present.
Response: Thanks for this insightful comment. In our study, the peptide sequence YYP was derived from the fraction less than <3 KD of our optimized enzymatic product (APGP-III), and the proportion of this fraction was high at 49.07%. Moreover, the amino acid composition of APGP, which is also the premise underlying our screening of tyrosinase inhibitory peptides. The title of our paper, “Optimization of a novel tyrosinase inhibitory peptide from Atrina pectinata mantle and its molecular inhibitory mechanism”, has two meanings: one is the optimization of the tyrosinase inhibitory peptide process, and the other is the determination of the inhibitory peptide's mechanism of action. Therefore, it was unanimously decided after our discussion that these two parts should be retained.
#Responses: Reviewer:
- The author's affiliation #1 seems curious. How many laboratories are there?
Response 1: The other four : National Research and Development Branch Center for Shellfish Processing (Zhanjiang); Guangdong Provincial Key Laboratory of Aquatic Products Processing and Safety; Guangdong Provincial Engineering Technology Research Center of Seafood; Guangdong Province Engineering Laboratory for Marine Biological Products, are research platforms of Guangdong Ocean University (College of Food Science and Technology).
(1) P1L7, I think "the tyrosinase inhibition activity" more better. What is APGP? Please define the term.
(2) Lineweaver-Burk analysis often reveals the maximum activity and half-maximum values but it could not realized the molecular-molecular interaction force. Please rephrase the last 4 lines.
Response 2: Thanks for Reviewer’s kind comments.
- Following your suggestion, we have changed “the tyrosinase inhibition” to “the tyrosinase inhibition activity” and redefined the term APGP to “Atrina pectinata mantle gelatin peptide (APGP)”.
- We have rephrased the last four lines to describe the results of molecular docking and Lineweaver-Burk analysis, respectively. The revised details can be found in the new revision (Line 11-14, revision).
- L2: Use ultraviolet here before page2-line1.
- Second paragraph line4, “tric acid” means “citric acid”?
- 3rd paragraph line4, It’s preferred that “ACE” first define the term then abbreviate as ACE. Line8-9. Did Gelatin peptide from fish scales show 61.7% inhibition or 61.7% inhibitory activity? What is 100% of tyrosinase inhibitory activity? ∆MI probably melanin index but it is preferred full spell in first.
- The last paragraph is the first mention of Atrina pectinata, but there is no description of why the focus was on the mantle of Atrina pectinata. I would like to see a more detailed description of why Atrina pectinata and why the focus was on the Atrina pectinata mantle. Please explain why the tyrosinase inhibitory peptide was known to exist in the Atrina pectinata mantle in the first place.
(5) Finally, is it correct to conclude that the author is not talking about a peptide obtained from Atrina pectinata, but an oligopeptide sequence YYP that has tyrosinase inhibitory activity?
Response 3: Thanks very much for these good comments and your hard work.
- We use “ultraviolet (UV)” in page1-line3 and replace UV for page2-line1 with abbreviations.
- In the fourth line of the second paragraph, there was a misspelling of “citric acid”, which has been corrected to read “triglycerides”.
- We have added definitions of ACE (“angiotensin converting enzyme”) and ΔMI (“skin melanin content ΔMI”) terms. We have replaced the sentence “A recent study showed that the gelatin peptide extracted from fish scales possess tyrosinase inhibitory activity of 61.7% (5 mg/mL)” with “A recent study showed that the gelatin peptide extracted from fish scales possess 61.7% tyrosinase inhibition (5 mg/mL)”.
- We have added the description of the economic value of the Atrina pectinata, and the low utilization of the mantle as a by-product thereof. The revised details can be found in the new revision (Penultimate line 5-11, Page 2, revision). In the existing studies, little research has been done on the bioactive peptides of the Atrina pectinata mantle, so we pre-determined the main nutrients and amino acid composition of the mantle. It was found that the dry matter content of the mantle was as high as 74.2±34 % protein, 20.9±0.45% ash and 4.7±0.00 % carbohydrate. The fat percentage was the lowest at 0.2±0.00 %, consistent with the high protein and low fat characteristics of shellfish. Table 1 shows the amino acid composition of the mantle (total amino acids 63.99±0.38 g/100g), of which the proportion of hydrophobic amino acids is about 34%. Effective tyrosinase-inhibiting peptides usually contain hydrophobic residues; therefore, we extracted gelatin from the mantle and measured its tyrosinase inhibition rate, confirming its tyrosinase-inhibiting effect.
Table 1. Amino acid composition of Atrina pectinata mantle.
|
Amino acid |
Content (g per 100 g) |
Amino acid ratio (%) |
Amino acid |
Content (g per 100 g) |
Amino acid ratio (%) |
|
Glu |
9.69±0.40 |
15.14±0.53 |
Thr |
2.96±0.02 |
4.63±0.01 |
|
Gly |
7.68±0.07 |
12.00±0.27 |
Val |
2.84±0.01 |
4.45±0.04 |
|
Asp |
6.76±0.06 |
10.49±0.04 |
Ser |
2.82±0.05 |
4.42±0.10 |
|
Arg |
5.32±0.04 |
8.31±0.02 |
Ile |
2.64±0.02 |
4.12±0.06 |
|
Leu |
3.52±0.08 |
6.93±0.10 |
Tyr |
2.38±0.04 |
3.72±0.09 |
|
Pro |
4.30±0.09 |
6.73±0.10 |
Phe |
2.35±0.07 |
3.67±0.13 |
|
Ala |
3.70±0.01 |
5.78±0.06 |
Met |
1.50±0.05 |
2.35±0.06 |
|
Lys |
3.41±0.00 |
5.33±0.03 |
His |
1.23±0.11 |
1.92±0.19 |
- We are very sorry for this negligence, it has already been revised. The aim of our study were to discover new potentially effective tyrosinase inhibitory peptides from the Atrina pectinata mantle, and then further explore the molecular mechanism of action of the inhibitory peptides, with a view to providing a certain basis for the subsequent application of Atrina pectinata mantle in whitening. Therefore, we have modified the last two sentences, as “Solid-phase synthesis of the screened inhibitory peptides and determination of IC50 values were performed, and the molecular inhibitory mechanism of optimal tyrosinase inhibitory peptide was investigated for the rational utilization of the Atrina pectinata Expanding its use as a natural skin whitener in the biomedical and cosmetic industries”.
(1) Line1, the laboratory is which one? If “the laboratory” prepared the gelatin previously, author should state the references. Probably, “tyrosinase.”
(2) What is enzymolysis? How many and how much protease were used at which pH and what temperature? Too little information on proteolytic conditions. 12000 r/min => Show as a gravitational acceleration, or at least rpm.
(3) What is RSM? Line5, What is 0.1 M PBS? Where did the “As” go? Is equation of (1) correct? Do A4 and A3 not subtract A1? What is mean “hole”? Did you analyzed these absorbance using 96-well plate?
(4) Table 1, The column Reagent/µL, separate Reagents and solution volume unit. The volume unit put the column solvents (T1-T4).
(5) 2.7, 2.8, If author want to use abbreviation of MW, use the MW instead of molecular weight there after.
(6) 2.7, Line6, missing parentheses “PEG”.
(7) 2.8, Line1, What is MWCO? State correctly, probably molecular weight cut off.
Response 4: Thanks for Reviewer’s kind comments.
- We have added a description of the source of the gelatin as follows : “ Atrina pectinata was purchased from the local Dongfeng Market in Zhanjiang, China, and the methods of gelatin from mantle (APG) preparation referenced to wang et al [22].”
- We apologize for this oversight and have added a description of the enzymatic one-factor process at 2.2 (Line1-5, revision). Moreover, we've changed “12,000 r/min” to “12,000 rpm/min”.
- “RSM” stands for “Response Surface Methodology” and “0.1 M PBS” is “0.1 mol/L Phosphate Buffer Solution”, we have added its full name to the revised manuscript. “As” was intended to represent the absorbance of the sample, but it was already stated otherwise in Table 1 (A1-A4), so we delete it in the revision. Equation of (1) is Referring to the Yu [23], both A4 and A3 contain the background reaction absorbance value of the enzyme solvent in A1, therefore, here we choose not to subtract A1. We used a 96-well plate for spiking and measuring absorbance. We apologize for the inaccuracy of the expression "hole" and have changed it to “well”. Please review at 2.4 in revision.
- We apologize for not noticing the reagents and solution volume unit error issue, it has now been fixed. The revised details can be found in Table 1 in the new revision.
- We use the abbreviation MW instead of molecular weight in both 2.7 and 2.8. Those appearing below are also synchronized and replaced with the corresponding abbreviations(3.4, 3.5, revision).
- We have added the parentheses of PEG parentheses (Line 6, 2.7, revision).
- It has been revised as “molecular weight cut off (MWCO)”(Line 1,8, revision).
(1) P9, 3.4, Line13-15 and Table4, This is my suggestion. Since Table 4 shows amino acid content, how about adding amino acid ratio information for each amino acid in Table 4, as in line 15?
(2) Is the term “enzymatic peptide” means enzymatic-digested peptide? Or a peptide has enzymatic activity?
(3) P10, line2-12 and Table5, Why don’t author display these textual information as as figures (plot graph)? I would think that would give clearer and more informative results. Moreover, I would like to see the elution profile of APGP along with the molecular standard profile.
(4) Table 6 title indicated unfractionated APGP but table 6 seems only fractionated APGP data. The data missing? In footnote, “Std. Deviation” should be “standard deviation”.
(5) P11, 3.6.1 regarding molecular docking section, Are the 10 peptides synthetic or textual sequences? If those peptides were synthesized, table 7 was in vitro results. But textual sequences, the results mean in silico?
(6) P13, Scientific names of plants should be italicized.
Response 5: Thanks for these helpful comments.
(1) We have added the amino acid ratio information in Table 4 of the revised manuscript.
(2) The term “enzymatic peptide” refers to the APGP obtained from Material 2.3 and the last phrase on page 9 has been corrected to read “The percentage of the above functional amino acids in the amino acid composition of the APGP was as high as 42.33 %, indicating that APGP is promising for the study of tyrosinase inhibitory activity”.
(3) We added the elution profile of APGP and labeled the graph with the peak times of the molecular standards (Figure 4, revision).
(4) We are very sorry for this error, the unfractionated APGP data (IC50 of 3.268± 0.048) has been inserted into the table 6 and the data re-analyzed for significance (Table 6, revision).
(5) In this study, we used multiple bioinformatics platforms based on peptide sequence scores and molecular weight from mass spectrometry to learn more information, such as toxicity prediction and biochemical property, prior to activity validation, and screened a total of eight eligible peptide sequences (Peptide sequences with MD <1 kDa and score >100,non-toxic with GRAVY value < 0). Therefore, we corrected the ten peptide sequences in Table 7 to eight peptide sequences, and the title of Table 7 was changed to “Potential tyrosinase-inhibiting peptide sequences: mass spectrometry and in silico screening.”
(6) We have added the scientific names of plants “Valerian tannins (Quercus acutissima Carr)” and “Apios isoflavone glucoside (Apios americana)”. In addition, italicizing the plants “Dalbergia ecastaphyllum” and “Prunus persica”. (3.6.2, revision)
We have tried our best to revise our manuscript overall according to all the comments above. On behalf of my co-authors, we thanks again for giving us so much kind comments and suggestions.
Round 2
Reviewer 1 Report
Comments and Suggestions for Authors
It is interesting topic and the manuscript is well organized. However, there still have some issues need to check.
The 3.6.1. Molecular Docking. The specific and detailed information is needed when comparing the Molecular Docking. Please refer this reference (Food Bioscience, 43(2021), 101313).
The reference should be updated in recent years with supplement volume number and page number.
Comments on the Quality of English Language
It is interesting topic and the manuscript is well organized. However, there still have some issues need to check.
The 3.6.1. Molecular Docking. The specific and detailed information is needed when comparing the Molecular Docking. Please refer this reference (Food Bioscience, 43(2021), 101313).
The reference should be updated in recent years with supplement volume number and page number.
Author Response
Dear Reviewer,
We have uploaded the "Response to Reviewer Comments" for the paper. Please see the attachment.
Kind regards,
Wen Wang
E-Mail:wangwen31@stu.gdou.edu.cn
Response to Reviewer Comments
We appreciate very much for Reviewer’s positive and constructive comments and suggestions on our manuscript. The responses are provided as following:
#Responses: Reviewer:
1- The 3.6.1. Molecular Docking. The specific and detailed information is needed when comparing the Molecular Docking. Please refer this reference (Food Bioscience, 43(2021), 101313).
Response 1: Thanks for Reviewer’s kind comments. We have carefully read the references you provided , with reference to their descriptive analysis of the docking results, and have added a description of the tyrosinase and its docking site in 3.6.1 as follows “The crystal structure of Agaricus bisporus tyrosinase (2Y9X) is in the form of a homotetramer, in which the active site consists of two copper atoms and six histidine residues in coordination (His61, His85, His94, His259, His292, His296) ”. In addition, detailed comparisons of the docking results for the four peptide sequences have been added, as described in the blue section of the revised manuscript (3.6.1, revision).
- The reference should be updated in recent years with supplement volume number and page number.
Response 2: We have updated seven new references in recent years, with the number of references [15, 18, 27, 29, 30, 41, 53], and added the volume and page numbers of reference 37. We have adjusted and modified the content according to the updated references 18 and 30 as follows: “Li [18] demonstrated that silver carp scale collagen peptides (SCPs1) can reduce tyrosinase activity and reactive oxygen species (ROS) content of B16F10 mouse melanoma cells, thus affecting melanogenesis, and is expected to be applied to skin whitening products.” and “The pH values affect substrates and enzymes by altering the charge distribution of proteins, as well as being able to cause changes in the exposure of the active site of enzymes [30].”
Reviewer 2 Report
Comments and Suggestions for Authors
I think we carefully improved what we pointed out.
A few more additional comments: in 2.2, the condition settings are described as A enzyme concentration, B pH, C time, and D reaction temperature, but in 2.3 they do not appear to match those conditions. Is the enzyme concentration not included in the study material?
Author Response
Dear Reviewer,
We have uploaded the "Response to Reviewer Comments" for the paper. Please see the attachment.
Kind regards,
Wen Wang
E-Mail:wangwen31@stu.gdou.edu.cn
Response to Reviewer Comments
We appreciate very much for Reviewer’s positive and constructive comments and suggestions on our manuscript. The responses are provided as following:
#Responses: Reviewer:
1- A few more additional comments: in 2.2, the condition settings are described as A enzyme concentration, B pH, C time, and D reaction temperature, but in 2.3 they do not appear to match those conditions. Is the enzyme concentration not included in the study material?
Response 1: We are very sorry for this negligence, in 2.2, we used a one-way experiment to explore the effects of different enzymatic-digested factors, including enzyme concentration, pH, time and reaction temperature, on tyrosinase inhibition rate and DH values. While in 2.3, we designed a three-factor, three-level response surface to optimize the enzymatic conditions using enzymatic temperature (A), pH (B), and time (C) as independent variables, without the independent variable of enzyme concentration. In order to distinguish the conditions in 2.2 and 2.3, we changed the sentence “ The effects of enzyme concentration (A, 2, 3, 4, 5, 6 U/mg), enzymatic pH (B, 5, 6, 7, 8), time (C, 3, 3.5 ,4, 4.5 h) and temperature (D, 45, 55, 65, 75 °C) of this protease on the inhibition of tyrosinase and degree of hydrolysis (DH) were explored.” in 2.2 to “The effects of enzyme concentration (2, 3, 4, 5 and 6 U/mg), enzymatic pH (5, 6, 7 and 8), time (3, 3.5 ,4 and 4.5 h) and temperature (45, 55, 65 and 75 °C) of this protease on the inhibition of tyrosinase and degree of hydrolysis (DH) were explored. ”